# Heavy Metal Accumulation in Common Aquatic Plants in Rivers and Lakes in the Taihu Basin

**DOI:** 10.3390/ijerph15122857

**Published:** 2018-12-14

**Authors:** Li Bai, Xiao-Long Liu, Jian Hu, Jun Li, Zhong-Liang Wang, Guilin Han, Si-Liang Li, Cong-Qiang Liu

**Affiliations:** 1Tianjin Key Laboratory of Water Resources and Environment, Tianjin Normal University, Tianjin 300387, China; baili@tjnu.edu.cn (L.B.); lijun5931@163.com (J.L.); zhongliang_wang@163.com (Z.-L.W.); 2School of Geographic and Environmental Sciences, Tianjin Normal University, Tianjin 300387, China; 3The State Key Laboratory of Urban and Regional Ecology, Research Center for Eco-Environmental Sciences, Chinese Academy of Sciences, Beijing 100085, China; hujian@vip.skleg.cn; 4School of Scientific Research, China University of Geosciences (Beijing), Beijing 100083, China; hanguilin@cugb.edu.cn; 5Institute of Surface-Earth System Science, Tianjin University, Tianjin 300072, China; siliang.li@tju.edu.cn (S.-L.L.); liucongqiang@vip.skleg.cn (C.-Q.L.); 6The State Key Laboratory of Environmental Geochemistry, Institute of Geochemistry, Chinese Academy of Sciences, Guiyang 550002, China

**Keywords:** aquatic plants, heavy metals, Taihu Lake, bioaccumulation capability

## Abstract

We investigated the concentrations of 10 heavy metals in *Potamogeton malaianus*, *Nymphoides peltata*, *Eichhornia crassipes*, and *Hydrilla verticillata* to evaluate their potential to bioaccumulate heavy metals and related influencing factors in Taihu Lake. Enrichment factor (EF) values of Cu, Cr, Mn, Ni, Zn, Co, Pb, and V were above 2.0, indicating moderate to significant contamination in sediment. Most of Ti, V, Cr, Mn, and Ni in *P. malaianus*, *E. crassipes*, and *H. verticillata* and V in *N. peltata* were within excess/toxic level in plants, but higher than normal level. Even though no aquatic plants in this study were identified as a hyperaccumulator, relatively higher concentrations in aquatic plants were found in Taihu Lake than have been found in other previous studies. Heavy metal in submerged plants, especially in their stems, seemed to be more closely related to metals in water and sediment than those in floating-leaf plants. Ratios of metals in stem versus leaves in all plants ranged from 0.2 to 25.8, indicating various accumulation capabilities of plant organs. These findings contribute to the application of submerged aquatic plants to heavy metal removal from moderately contaminated lakes.

## 1. Introduction

Heavy metals in aquatic environments, such as lakes and rivers, have been studied extensively because of their toxicity, persistence, and tendency to bioaccumulate. Many studies have shown that aquatic plants are sinks for heavy metals in aquatic ecosystems ([1], and that different aquatic plant species accumulate variable amounts of different heavy metals. For example, studies have shown that floating plants such as water hyacinth (*Eichhornia crassipes*) [2], emergent plants such as cattail (*Typha. latifolia*) [3], and submerged plants such as *Hydrilla verticillata* [4,5], *Ceratophyllum demersum* [5], and *Potamogeton malaianus* [6] have significant capacities to accumulate heavy metals. Because of their capacity to accumulate heavy metals, several species of aquatic macrophytes such as *H. verticillata* and water hyacinth have been used to remove heavy metals from waste water. The accumulation capabilities of aquatic plants generally decrease from submerged plants to floating plants and then to emergent plants, but this in turn is influenced by the plant species and aquatic environment [7].

Eventually, the metal concentrations in water reflect the current condition, whereas surface sediment is an important reservoir of heavy metals in the aquatic environment. Aquatic plants can uptake large amounts of metals from water and/or sediment through active and passive absorption, with this absorption capacity of metals through different organs such as roots, stems, and leaves, making these plants suitable for heavy metal alterations in the aquatic environment [7,8]. However, there is still a lack of information about the capability of various aquatic plants to accumulate heavy metals and the relationships between the plants and the metal concentrations in water and sediment [1,9]. More knowledge about the accumulation capacity of various aquatic plants is needed from direct observations. Moreover, measuring the accumulation of heavy metals in aquatic plants can provide time-integrated information about the presence of metals in the aquatic ecosystems.

Taihu Lake continuously received industrial waste and domestic sewages. Researchers have examined the metal contamination in the lake bottom sediments and suspended particulate matter [10,11], but little is known about heavy metal uptake by aquatic plants. Macrophytes dominated in Xukou Bay since the 1990s, and the high metal concentrations in sediment motivated us to study the accumulation capacity of heavy metals by aquatic plants in lake and rivers in Taihu. Here, we compared the heavy metal bioaccumulation in submerged macrophytes and floating plants. The aims of our study were to (i) examine the heavy metal concentration variation in waters, sediments, and different plant species in the Taihu Lake system; (ii) assess the potential ability of different organs of aquatic plants to accumulate heavy metals; and (iii) assess the effects of sediment and water on the heavy metal accumulation of aquatic plants.

## 2. Methods and Materials

### 2.1. Description of Research Area

Taihu Lake, located in the southeastern part of the lower reaches of the Yangtze River, is the third largest freshwater lake in China. Despite its total surface area of 2338 km^2^, Taihu Lake has a mean water depth of only 1.9 m [11]. There are numerous canals, rivers, and streams (over 100) around Taihu Lake. Due to rapid economic growth, serious water contamination has been experienced in this lake since the 1980s. The pollution is most serious in the northern part of the lake [10,12], where highly concentrated industrial and domestic sewages are discharged, thus the aquatic plant growth is limited in that area.

### 2.2. Collection and Analysis of Samples

We collected samples of water and sediment from 10 river sites (surrounding rivers, SR) and 10 lake sites in October 2013 (Figure 1). Four of the most common aquatic plant species, namely *P. malaianus*, *N. peltata*, *E. crassipes*, and *H. verticillata*, were collected from Xukou Bay (marked as XKW), Eastern Taihu Lake (marked as DTH), Western Taihu Lake (marked as XTH), and Taipu River (marked as TPH) (Figure 1). To facilitate comparison, two additional sampling sites without aquatic plants were chosen in the central lake area (HX-1 and HX-2).

During sample collection, water samples were immediately transported to the laboratory and then were stored in the dark at 4 °C for preliminary laboratory handling. Sediment samples were stored in 50-mL centrifuge tubes and were frozen and stored in the laboratory. Because of different spatial distributions, some of the plants were not available at all sites; for example, *N. peltata* was only found in the eastern part of Taihu (DTH in Figure 1) and Xukou Bay (XKW in Figure 1). We collected three samples of each plant species as replicates. Plant samples were collected from the root and then were washed with deionized water, placed in polyethylene bags, and transported back to the laboratory in coolers.

Water samples were filtered through a 0.45-μm cellulose acetate membrane by glass vacuum filter. Samples of sediment and aquatic plants were freeze-dried at −50 °C for 48 h and then ground through a 100-mesh sieve. All containers were washed with acid before using to remove potential background contamination. Sediment and plant samples were microwave digested with HCl-HNO_3_-HF-HClO_4_, and HCl-HNO_3_-HClO_4_, respectively, using a microwave accelerated reaction system (Mars 5, CEM Corporation, Matthews, NC, USA). Heavy metals were detected using an inductively coupled plasma mass spectrometer (ICP-MS; Perkin-Elmer SCIEX Elan 9000, PerkinElmer, Inc., Shelton, CT, USA) equipped with a Cross-flow Ryton Nebulizer (PerkinElmer, Inc., Shelton, CT, USA). The relative error for all metals was less than 8% and the recoveries of the reference materials (GBW-07310, National Standard Material Research Center, Beijing, China) were between 89–105%.

### 2.3. Sediment Quality Assessment

To assess and quantify the impact of human activities on heavy metals, we calculated enrichment factors (EFs). Mathematically, EF is expressed as:EF = (M/X)_sample_/(M/X)_background_
where M is the evaluated element, X is the reference element, and (M/X)_sample_ and (M/X)_background_ are the ratios of an evaluated element and the reference element in the examined and geochemical background, respectively. As a geochemical common factor, aluminum has often been used as the reference element [13]. Thus, the EFs were calculated using Al as reference element and heavy metal background values of soils in Jiangsu Province [14] were chosen as the background. An EF < 2 represents no contamination to minor contamination, 2–5 represents moderate contamination, 5–20 represents significant contamination, 20–40 represents very high contamination, and >40 represents extremely high contamination [10,13].

### 2.4. Statistical Analyses

Pearson correlation coefficients were calculated to assess the relationship between metal concentrations in plants with that in water and sediments. The analysis of variance (ANOVA) with Tukey’s Honestly Significant Difference (HSD) test was used to determine differences in element concentration between waters, sediments, and aquatic plants. A *t* test was used to determine the significance of the differences between the concentration of elements in stems and leaves of plants. The probability level was set at 0.05. All statistical calculations were carried out using IBM SPSS Statistics 24 (International Business Machines Corporation, Armonk, NY, USA).

## 3. Results and Discussion

Concentrations of the metals in waters and sediments are summarized in Table 1. The mean concentration of Ni in surface water exceeded the limits of Grade II of the quality standard (GB3838-2002). Overall, the concentration of heavy metals in water in Taihu was low; Yu et al. (2012) [15] attributed this mainly to the higher dissolved organic carbon (DOC) in Taihu, and Zhu et al. (2005) [16] specified this as suspended solids in the water, as metals are easily sequestered or combined with organic carbon and then settle to the bottom.

In the sediment, there was a wide variation in the contents of heavy metals. Most of the heavy metals in sediment had higher concentrations than the background values in local soils, except Ti and Fe. Additionally, most of the heavy metal concentrations were higher than those in China crust as well, except Fe. Heavy metals in waters and sediments from different lake areas also showed obvious differences (Figure 2). Mn, Ni, and Zn in TPH River and surrounding rivers (SR) were higher than lake waters. Higher concentrations of Ni, V, and Ti in the sediments of the central lake area (HXQ) in comparison to other lake areas may be related to industrial wastewater input. Most of the rivers flowing into west and north bank of Taihu Lake received a large amount of wastewater from industries each year, resulting in pollutants accumulating in HXQ [15]. With the exception of the Fe concentrations, which have not varied over the past 15 years, sediment concentrations of all heavy metals have increased noticeably since 2001 [12]. In particular, concentrations of Cu, Pb, Mn, and Cr have increased noticeably in sediment in Xukou Bay and Western Taihu Lake relative to the results reported by Liu et al. (2012) [10]. The concentrations of Fe and Cu in sediment were highest in Xukou Bay in comparison to the other lake areas and rivers, with averages of 33,256 ± 3045 and 67.1 ± 14.0 mg kg^−1^, respectively. Thus, despite the numerous regulations and policies that have been enacted in recent years in an effort to reduce the pollutants in Taihu, in addition to projects management efforts, such as mud-cleaning engineering, this study revealed that further efforts are need to reduce heavy metals in waters and sediments.

Calculated EF values for heavy metals in sediments are shown in Figure 3. EF values of Fe and Ti in sediment presented closely to 1, indicating that these heavy metals originate mainly from natural lithogenic sources and are not influenced by human contamination, which supports the previous finding by Liu et al., 2012 [10]. The EF ranges of Cu, Cr, Mn, Ni, Zn, and Co were all between 2 to 5, suggesting moderate contamination from human activities in all lakes and rivers. The EF ranges of Pb and V in DTH and XTH were larger than 5, thus showing significantly contamination in the sediment. In addition, the EFs of Ni, Zn, and Mn were higher in the sediments of the TPH and surrounding rivers than those in the lacustrine sediments. This indicates that pollutants carried by the rivers are probably the main source of heavy metal accumulation in the lake ecosystem and potentially pose considerable ecological risks [11], because in shallow lakes such as Taihu, those fluvial sedimentary pollutants are more likely to be resuspended and cause secondary contamination to the water environment [15].

As shown in Table 2 and Table 3, all heavy metals were detected in four aquatic plants. Fe, Ti, Cr, Cu, and Pb showed highest values in *P. malaianus* (*p* < 0.05), while Mn, Ni, and Zn were highest in *E. crassipes* (*p* < 0.05). In addition, high concentrations of several heavy metals, such as Fe, Mn, Ti, V, Zn, and Pb, were also observed in *H. verticillata.* Th Co, Cu, Zn and Pb concentrations determined in four aquatic plants from Taihu Lake were mostly within the normal concentration ranges for plant leaves given by Kabata-Pendias (2010) [18], which were not consistent with their enrichments (EFs > 2.0) in sediments. This inconformity showed that Co, Cu, Zn, and Pb contaminants in sediment have not resulted in corresponding metal accumulation in aquatic plants. Most Ti, V, Cr, Mn, and Ni concentrations in *P. malaianus*, *E. crassipes*, *H. verticillata*, and V in *N. peltata* were within levels considered to be toxic in plants. According to the threshold values that were used to define hyperaccumulators by Xing et al. 2013 [1], no heavy metal concentrations observed in this study exceed the accumulation threshold (Table 2), although the concentrations and percentages in aquatic plants in Taihu Lake documented in this study are higher than other previous studies. Cr and Mn concentrations in *P. malaianus*, *E. crassipes*, and *H. verticillata*; Fe concentrations in *P. malaianus* and *H. verticillata*; and Ni concentrations in *P. malaianus* and *E. crassipes* were higher than those in other lakes (Table 2), which indicates that in consideration of their important role in the food chain of the lake ecosystem, the relatively high concentration of heavy metals in aquatic plants is a cause for concern. However, there was a wide variation in the contents of heavy metals in different aquatic plants among the different lake areas. The heavy metal concentrations in *N. peltata* were noticeably lower than in the other three plants. In fact, *N. peltata* has not attracted attention as a heavy metal accumulating plant before; we showed that it had a comparable ability to accumulate Fe as *E. crassipes.*

Concentrations of heavy metals, and in particular Fe, Cr, Pb, Co, and Ti, in *P. malaianus* and *H. verticillata* were generally higher than those in *E. crassipes* and *N. peltata* (*p* < 0.05) (Table 2 and Table 3). *P. malaianus* and *H. verticillata* are submerged aquatic macrophytes, while *E. crassipes* and *N. peltata* are floating-leaf macrophytes, so this result, which agrees with earlier conclusions of Wang et al. (2014) [9], supports the hypothesis that floating-leaf macrophytes have a lower accumulation capability than submerged macrophytes. The potential of *H. verticillata*, a rapid-growing, common aquatic plant with worldwide distribution, to accumulate heavy metals such as Pb, Hg, Cu, Cd, Cr, Ni, and As has already been demonstrated [4,5]. Xing et al. (2013) [1] reported that it was able to hyperaccumulate Fe. Xue et al. (2010) [4] reported that *H. verticillata* absorbed Cu and suggested that it could be used to remove Cu from moderately polluted waters. Even though *H. verticillata* was not found to be a hyperaccumulator in this study, concentrations of heavy metals that were comparably high as those observed in other lakes were documented. In addition, in this study, we also found that *E. crassipes* had a higher accumulation ability for Mn, Ni, and Cu than the other three plants in Taihu Lake (Table 2 and Table 3). Actually, *E. crassipes* is one of the most researched aquatic macrophytes for phytoremediation of both eutrophic waters and heavy metal-polluted waters [2]. In total, based on the increasing heavy metal contamination in Taihu Lake, the aquatic plants’ role in heavy metal remove and indication of pollution should attract more attention.

While the heavy metal accumulating ability of plant roots has already been studied extensively, we still have little information about the different accumulating abilities of the stems and leaves of aquatic plants [9]. The results from this study show that the different metals were present at different concentrations in the leaves and stems of aquatic plants. We used the ratio of metals in stems and leaves to compare differences. The ratio of metals in stem/leaves ranged from 0.2 to 1.3 in *P. malaianus*, 0.5 to 25.8 in *E. crassipes*, 0.8 to 2.2 in *N. peltata*, and 0.6 to 2.1 in *H. verticillata*. The ratio varied a lot mainly due to the same aquatic plants being collected from different lake areas. For example, with the exception of Cr, the metal concentrations in *H. verticillata* collected from Xukou Bay were higher in the leaves than in the stems (ratio of stem/leaves > 1.0), while there was a tendency of all the heavy metals to accumulate more in the stems than in the leaves of *H. verticillata* in the eastern part of the lake (Figure 4). Similarly, the Cr, Mn, Fe, Co, and Ni concentrations in *N. peltata* collected from sites DTH-2 and XKW-2 were higher in stems than in the leaves. Concentrations of Ti, V, Mn, Fe, Co, Zn, and Pb in *P. malaianus* from site XKW-1 were higher in stems than in leaves, while heavy metal concentrations were higher in leaves than in stem in XTH-1. On the whole, the accumulation ability of the leaves and stems of aquatic plant seems to be related to aquatic environment conditions, including the concentrations of heavy metals in water and sediment. Similar findings have previously been presented in studies on heavy metal concentrations in fishes [8]. Considered together with overall comparison in Table 1, even though submerged plants had stronger accumulation abilities than floating-leaf plants, further studies are required to better understand the mechanisms of heavy metal accumulation in the stems and leaves of plants, especially between different plant species.

Various positive correlations were found between aquatic plants, water, and sediment (Figure 5). The correlations between the metal contents in the stems of the submerged plants (*P. malaianus* and *H. verticillata*) and the sediments tended to be stronger than the correlations between the stems of floating plants and sediments. Heavy metal concentrations in submerged plants, and, in particular, the concentrations in stems, seemed to be more closely related to their concentrations in water and sediment than those in floating-leaf plants. In addition, metal concentrations in the leaves of floating-leaf plants were affected by their concentrations in water. These findings should be further explored in future studies assessing the accumulation ability of plants organs.

## 4. Conclusions

Accumulation of heavy metals in waters, sediments, and four common aquatic plants in rivers and lakes of the Taihu Lake area were investigated. Most of the heavy metal concentrations in sediment were higher than background values in local soils, except Ti and Fe. Additionally, most of the heavy metal concentrations were higher than those in China crust as well, except Fe. Heavy metals in waters and sediments from different lake areas also showed obvious differences. EF values in the sediment indicated that most of the heavy metal concentrations, except Fe and Ti, were represented moderate to significant contamination levels resulting from human activities in the regional lakes and rivers. All heavy metals were detected in the four aquatic plants. Co, Cu, Zn, and Pb concentrations in the four aquatic plants from Taihu Lake were mostly within the ranges of normal concentration values for plant leaves, while most of the Ti, V, Cr, Mn, and Ni concentrations in *P. malaianus*, *E. crassipes*, *H. verticillata*, and V in *N. peltata* were within excess/toxic levels. Some obvious differences were observed in the heavy metal concentrations in the stems and leaves of the aquatic plants, suggesting that the accumulation capacity of the different organs varied a lot. This study also found that heavy metal concentrations in water and sediment appear to affect the heavy metal accumulation ability of plants to some extent. Even though no heavy metal concentrations observed in this study exceed the accumulation threshold, given the higher concentrations and percentages of heavy metals observed in aquatic plants in Taihu Lake in this study in comparison to other studies, and considering the important role that aquatic plants have in the food chain of lake ecosystems, the relatively high levels of heavy metals in the aquatic plants documented here present a cause for concern that necessitates further attention.

## Figures and Tables

**Figure 1 ijerph-15-02857-f001:**
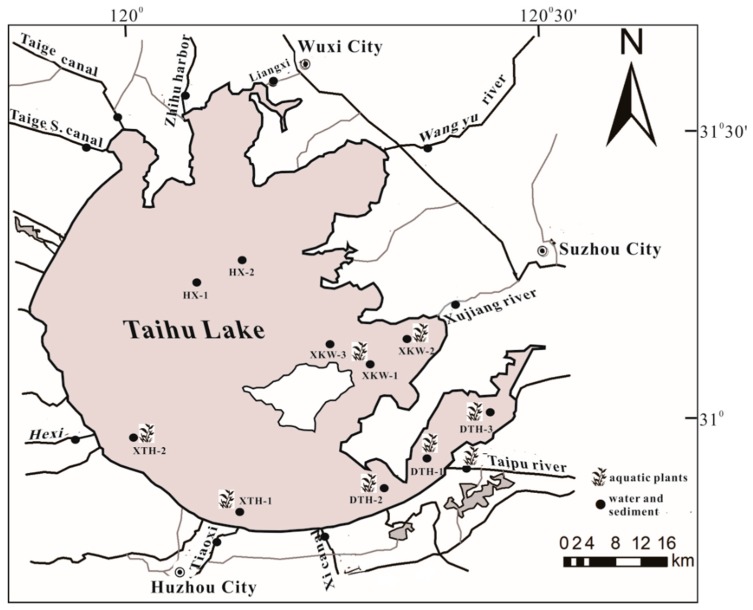
Sampling sites in Taihu Lake and its surrounding rivers.

**Figure 2 ijerph-15-02857-f002:**
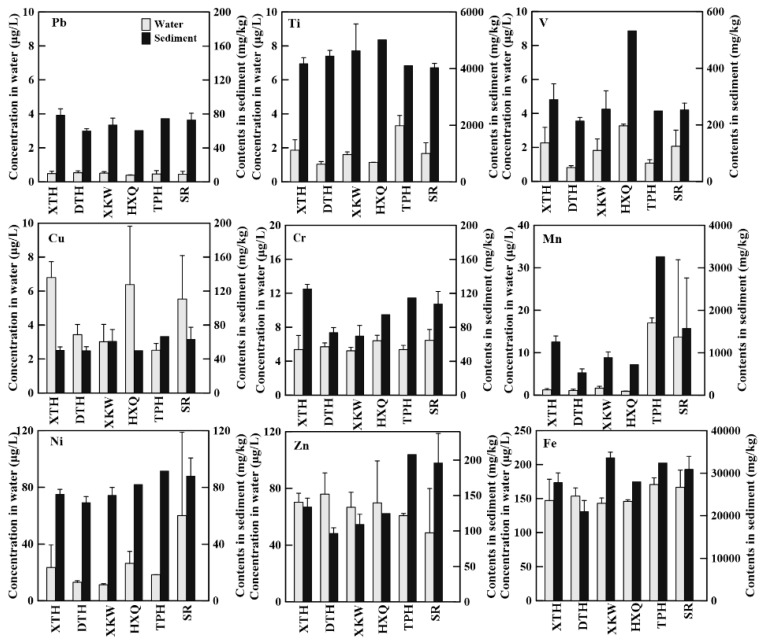
Heavy metal concentrations in water and surface sediments from Taihu Lake and rivers: Xukou bay (XKW), Western Taihu Lake (XTH), Eastern Taihu Lake (DTH), Central Taihu Lake (HXQ), Taipu River (TPH), Surrounding Rivers (SR), error bars represent SD. Captions are the same in the following figures.

**Figure 3 ijerph-15-02857-f003:**
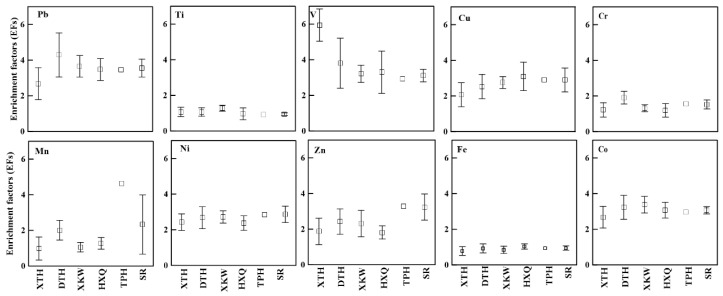
Enrichment factors (EFs) of heavy metals in sediments of Taihu Lake and rivers.

**Figure 4 ijerph-15-02857-f004:**
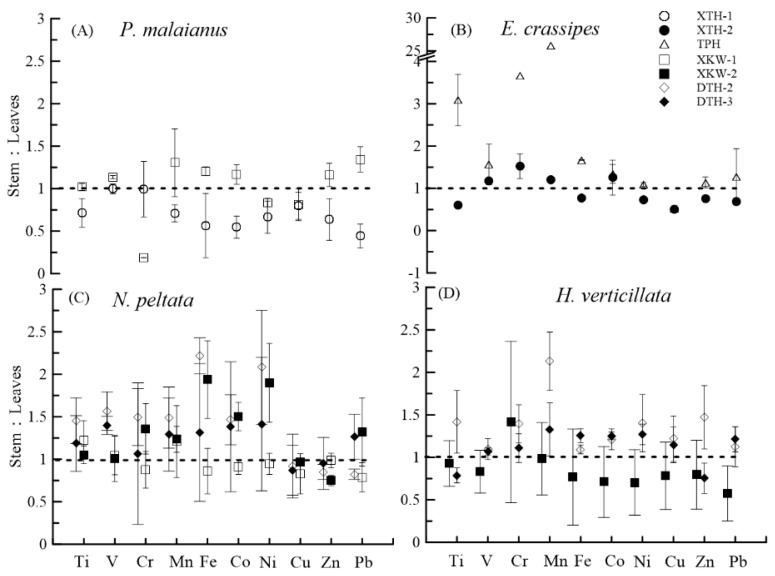
Heavy metal distribution in stems and leaves of different aquatic plants from different lake areas. Error bars represent the standard deviation (SD), dashed horizontal line represents the 1:1 ratio (stem: leaves) of metal concentrations quantified in plant tissues. (**A**) *Potamogeton malaianus*; (**B**) *Eichhornia crassipes*; (**C**) *Nymphoides peltata*; (**D**) *Hydrilla verticillata*.

**Figure 5 ijerph-15-02857-f005:**
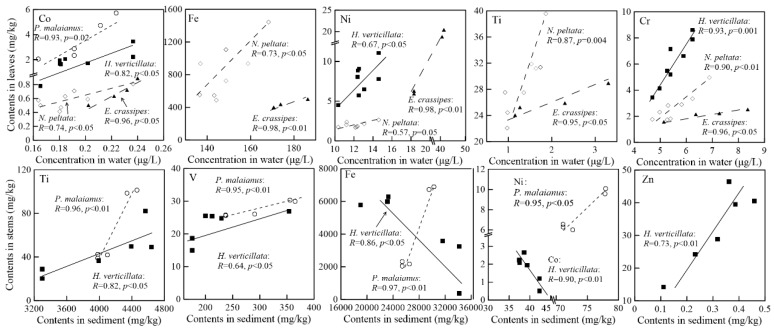
Correlations between heavy metals in water, sediment, and stems and leaves in plants. Pearson correlations are provided as lines and dashed lines. *R* and *p* values are also noted.

**Table 1 ijerph-15-02857-t001:** Heavy metal concentrations in surface waters (µg L^−1^) and sediments (mg kg^−1^, dry weighted (DW)) of Taihu Lake.

Element	Surface Water		Sediment		
	Mean ± SD (*n* = 20)	Environmental Quality Standard (Grade II) ^a^	Mean ± SD (*n* = 20)	Background Value ^b^	China Crust ^c^
Ti	1.7 ± 0.7	≤100	4400.4 ± 564.1	4564 ± 1376	660
V	1.7 ± 0.9	≤50	255.7 ± 54.6	87.0 ± 40.6	99
Cr	5.4 ± 1.0	≤50	92.4 ± 27.5	75.6 ± 6.0	63
Mn	2.4 ± 4.1	≤100	1076.0 ± 679.0	718 ± 548	780
Fe	149.1 ± 19.6	≤300	28,238 ± 5514	35,000 ± 12,000	50,800
Co	0.2 ± 0.1	≤1000	40.4 ± 2.4	14.6 ± 10.4	32
Ni	16.3 ± 10.2	≤20	74.4 ± 6.7	32.8 ± 18.0	57
Cu	4.4 ± 2.0	≤1000	54.9 ± 10.0	23.4 ± 14.4	38
Zn	69.9 ± 10.6	≤1000	120.7 ± 30.6	64.8 ± 30.2	86
Pb	0.5 ± 0.1	≤10	69.6 ± 9.7	22.0 ± 12.0	15

^a^ The environmental quality standards for surface water in China (GB3838-2002). ^b^ Heavy metal background values of soils in Jiangsu Province [14]. ^c^ From Institute of Geochemistry, Chinese Academy of Science (IGCAS) [17]. SD represents standard deviations.

**Table 2 ijerph-15-02857-t002:** Concentration and accumulation of heavy metals in aquatic plants in Taihu Lake.

Heavy Metals	Aquatic Plants	Range in this Study (mg kg^−1^ DW)	Normal Level in Plant Leaves ^a^	Excess/Toxic Level in Plants ^a^	Threshold (%) ^b^	Percentage in This Study	Percentage in Literature
Ti	*Potamogeton malaianus*	33.8~102.1	—	50–200	0.1	0.0102	—
	*Eichhornia crassipes*	14.8~76.7				0.0077	—
	*Nymphoides peltata*	22.1~56.1				0.0056	—
	*Hydrilla verticillata*	17.5~82.2				0.0082	—
V	*Potamogeton malaianus*	24.2~30	0.2–1.5	5–10	0.1	0.0030	—
	*Eichhornia crassipes*	16.2~30.9				0.0031	—
	*Nymphoides peltata*	14.8~27.4				0.0027	—
	*Hydrilla verticillata*	11.6~26.9				0.0027	—
Cr	*Potamogeton malaianus*	4.2~44.2	0.1–0.5	5–30	0.1	0.0044	≈0.0016 ^e^
	*Eichhornia crassipes*	1.5~8.4				0.0008	0.0001 ^f^
	*Nymphoides peltata*	1.7~5.5				0.0006	—
	*Hydrilla verticillata*	3.4~10.4				0.0010	≈0.0005 ^e^
Mn	*Potamogeton malaianus*	151.5~938.1	30–300	400–1000	1	0.0938	0.0304–1.9000 ^c^
	*Eichhornia crassipes*	145.1~4093.4				0.4093	0.12 ^f^
	*Nymphoides peltata*	42.1~386.4				0.0386	0.0536–0.0792 ^d^
	*Hydrilla verticillata*	823.7~3041.8				0.3042	≈0.0900 ^e^
Fe	*Potamogeton malaianus*	2038.8~6896.2	—	—	1	0.6896	≈0.3000 ^e^
	*Eichhornia crassipes*	349.3~3613.5				0.3614	—
	*Nymphoides peltata*	485.9~2722.2				0.2722	—
	*Hydrilla verticillata*	366.4~6500.7				0.6501	≈0.1250 ^e^
Co	*Potamogeton malaianus*	1.7~5.7	0.02–1	15–50	0.1	0.0006	≈0.0003 ^e^
	*Eichhornia crassipes*	0.5~2.7				0.0003	0.0001 ^f^
	*Nymphoides peltata*	0.4~1.6				0.0002	—
	*Hydrilla verticillata*	0.5~3.4				0.0003	≈0.0002 ^e^
Ni	*Potamogeton malaianus*	6~12.6	0.1–5	10–100	0.1	0.0013	≈0.0010 ^e^
	*Eichhornia crassipes*	6.2~20.3				0.0020	0.0011 ^f^
	*Nymphoides peltata*	1.6~5.2				0.0005	0.0003–0.0009 ^d^
	*Hydrilla verticillata*	2.1~11.1				0.0011	0.0020 ^g^
Cu	*Potamogeton malaianus*	5.7~16.1	5–30	20–100	0.1	0.0016	0.0036-0.0093 ^c^
	*Eichhornia crassipes*	4.8~21.6				0.0022	0.0008 ^f^
	*Nymphoides peltata*	2.2~7.5				0.0008	0.0004–0.0007 ^d^
	*Hydrilla verticillata*	2.4~7.9				0.0008	0.0152 ^g^
Zn	*Potamogeton malaianus*	21.7~75.2	27–150	100–400	1	0.0075	0.0106–0.7190 ^c^
	*Eichhornia crassipes*	41.3~75.3				0.0075	0.0019 ^f^
	*Nymphoides peltata*	15.7~37.1				0.0037	0.0038 ^d^
	*Hydrilla verticillata*	14.2~46.5				0.0046	0.0057 ^g^
Pb	*Potamogeton malaianus*	3.1~11.3	5–10	30–300	0.1	0.0011	0.0108–0.0604 ^c^
	*Eichhornia crassipes*	1.4~6.7				0.0007	0.00002 ^f^
	*Nymphoides peltata*	1.7~4				0.0004	0.0004–0.0010 ^d^
	*Hydrilla verticillata*	1.4~10.4				0.0010	0.0090 ^g^

“—” means no data or reference available. Percentages were calculated according to Xing et al. 2013 [1] with transforming the maximum concentration from unit of mg kg^−1^ into %; Data from ^a^ [18]; ^b^ [1]; ^c^ [6]; ^d^ [19]; ^e^ [20]; ^f^ [21]; ^g^ [22]. DW: dry weight.

**Table 3 ijerph-15-02857-t003:** Heavy metal concentrations in leaves and stems of the chosen aquatic plants (mg kg^−1^, DW).

Heavy Metals	*Potamogeton malaianus*		*Eichhornia crassipes*		*Nymphoides peltata*		*Hydrilla verticillata*	
	Leaves	Stem	Leaves	Stem	Leaves	Stem	Leaves	Stem
Ti	73.5 ± 26.5	54.1 ± 31.7	25.4 ± 3.1	35.7 ± 35.5	29.4 ± 5.4	35.8 ± 9.0	50.1 ± 17.9	44.7 ± 19.9
V	25.9 ± 1 *	26.6 ± 2.3 *	19.5 ± 4.0	24.6 ± 5.5	17.8 ± 3.2	21.6 ± 3.6	22.7 ± 4.8 *	22.7 ± 4.4 *
Cr	23.2 ± 18.8	9.3 ± 3.3	2.1 ± 0.4	4.7 ± 3.2	2.7 ± 1.1 *	2.9 ± 1.0 *	6.1 ± 1.8	7.0 ± 2.6
Mn	561.0 ± 353.0	481.0 ± 162.0	474.0 ± 363.0	1994.0 ± 1817.0	167.0 ± 71.0	220.0 ± 120.0	1268.0 ± 314.0	2040.0 ± 850.0
Fe	5207 ± 1770	3334 ± 2377	427.0 ± 48.0	456.0 ± 171.0	841.0 ± 329.0	1209.0 ± 584.0	4427 ± 1811	4464 ± 2188
Co	3.5 ± 1.6	2.4 ± 0.9	0.7 ± 0.2	0.9 ± 0.1	0.6 ± 0.1	0.8 ± 0.3	1.9 ± 0.8 *	1.9 ± 0.8 *
Ni	10.8 ± 2.3	7.2 ± 1.9	12.8 ± 7.80 *	11.7 ± 4.5 *	2.0 ± 0.3	3.1 ± 1.2	7.7 ± 2.1 *	7.7 ± 2.6 *
Cu	11.2 ± 4.0	9.9 ± 4.2	10.7 ± 1.70	5.6 ± 0.8	4.9 ± 1.5 *	4.4 ± 1.5 *	5.5 ± 1.5	6.3 ± 1.9
Zn	46.7 ± 24.9	35.0 ± 12.2	59.5 ± 17.2 *	54.0 ± 3.1 *	28.8 ± 4.9 *	26.5 ± 6.2 *	29.8 ± 12.4	32.1 ± 11.0
Pb	7.4 ± 3.2	4.5 ± 1.6	2.6 ± 0.8 *	2.2 ± 0.6 *	2.4 ± 0.4 *	2.4 ± 0.7 *	5.6 ± 2.4	5.2 ± 2.6

The average concentrations ± mean deviation are presented; * indicates similar values between stem and leaves of the same aquatic plant by *t*-test method (*p* < 0.05). DW: dry weight.

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
