# Peer review of "Heavy Metal Accumulation in Common Aquatic Plants in Rivers and Lakes in the Taihu Basin"

_ijerph, 2018, doi:10.3390/ijerph15122857_

Round 1
Reviewer 1 Report
The research topic is interesting for journal readers, but the manuscript needs further improvements before potential publication.
First, the aim should be evidenced in abstract and introduction sections.
Second, the methodological approach should be explained in a clearer way.
Third, the interpretation of results should be enriched also in terms of policy implications of the analysis.
Author Response
Response to Reviewer 1 Comments
Point 1: First, the aim should be evidenced in abstract and introduction sections.
Response 1: Thanks. We have changed and corrected the aims and the introduction sections, and compared them for consistency.
Point 2: Second, the methodological approach should be explained in a clearer way.
Response 2: We have classified the part of methodological approach into 4 parts of 2.1. Description of Research Area, 2.2. Collection and Analysis of Samples, 2.3. Sediment Quality Assessment, 2.4. Statistical Analyses for better understanding. And have revised the 2.1. Thanks for suggestion.
Point 3: Third, the interpretation of results should be enriched also in terms of policy implications of the analysis.
Response 3: By comparing the results in study to other studies, we may could talk about something in terms of policy implications. However, as we know, the heavy metal contamination is not easy to reduce because of their special chemical characteristics. So here we added some sentence to state the policy implication in the paragraph of stating the results of heavy metals in sediments and rivers. “Thus, despite of numerous regulations and policies have been enacted recent years on reducing the pollutants in Taihu, also including projects management such as mud-cleaning engineering, this study revealed that reducing heavy metals in waters and sediments still need further efforts. ” Thanks for suggestion.
Reviewer 2 Report
Dear authors
Please, find my suggestions and my comments in the pdf file of your manuscript.
Because some points remain unclear, I suggest reconsideration of your manuscript after correction.

Author Response
Response to Reviewer 2 Comments
Most of reviewer’s comments have been answered and corrected accordingly in the pdf MS as attached in the system. We chose some of the questions from reviewer and explained as follow. Thanks.
Point 1: In table 1, the authors should add the standard deviation for China Crust and soil background values.
Response 1: Metals concentrations from China Crust were cited from IGCAS 2000 Table 1.12 on page 43, we have checked the data sources and other literatures, no standard deviation provided. Thanks very much for advice. Standard deviations for soil background values have been added. Thanks.
Point 2: In table 1, why did the authors choose background values of soils.
Response 2: In some cases, the high heavy metal concentration in sediments may related to soil particulate matters because of the terrigenous detrital brought by the flood and riverine flow. Comparing the heavy metal in sediment and local soil background will do help to make clear whether these pollutants are from soil or from other sources such as industrial. thanks.
Point 3: The authors should consider local soils and after China crust. They should take into consideration the standard deviation if they have these data.
Response 3: Thanks, we have rewritten this paragraph, and compared the results separately with local soils and China Crust.
Point 4: The authors should also discuss about the concentrations of metals in water and those in sediments. If moderate contaminations of sediments have been highlighted by the authors, the metal concentrations in water are very low suggesting that the solubility of metals in sediments is very low.
Response 4: Thanks a lot for advices. We discussed about the low concentration in water, and added them into the first paragraph of the Discussion. "Overall, the concentration of heavy metals in water in Taihu was low, Yu et al. (2012)[13] attributed this mainly to the higher DOC in Taihu, and Zhu et al. (2005)[14] specified this as suspended solids in the water, as metals are easily sequestered or combined with organic carbon and then settle to the bottom.".
Point 5: The authors should detail this point. What is exactly the risk if the contamination level of water is low? Is there in the current study a problematic related to the reconversion of dregged sediments?
Response 5: Actually, this kind of risk mainly attributed to the re-suspended of particulate matters and sediments, which accumulated with high contents of heavy metals and will directly release the heavy metal into water or absorbed by aquatic plants. Hence, we added a sentence to the end of this paragraph for better understanding as follow: “because in shallow lakes such as Taihu, those fluvial sedimentary pollutants are more likely to be re-suspended and cause secondary contamination to the water environment”
Point 6: This sentence should be reformulated since this study not reported how metals accumulate in stems and leaves. There is no mechanistic study in the current paper.
Response 6: Yes, it is. We have changed this sentence into "While the heavy metal accumulating ability of plant roots has already been studied extensively, we still have little information about different accumulating ability in stems and leaves of aquatic plants "
Point 7: How this percentage was calculated
Response 7: Percentage were calculated according to Xing et al. 2013 with transforming the maximum concentration from unit of mg kg-1 into %. We have added this instruction in table caption.

Reviewer 3 Report
Thanks for the wonderful work you do. Good paper with some revisions needed. You will find my comments below.
Introduction:
Talk about why we care about heavy metals bioaccumulation in plants as related to human health. Focus on lead since it is the most studied one. Specifically, speak to its effects on numerous organ systems such as the;
1) cardiovascular system
Lanphear, Bruce P., Stephen Rauch, Peggy Auinger, Ryan W. Allen, and Richard W. Hornung. "Low-level lead exposure and mortality in US adults: a population-based cohort study." The Lancet Public Health 3, no. 4 (2018): e177-e184.
Obeng-Gyasi, Emmanuel, Rodrigo X. Armijos, M. Margaret Weigel, Gabriel M. Filippelli, and M. Aaron Sayegh. "Cardiovascular-Related Outcomes in US Adults Exposed to Lead." International journal of environmental research and public health 15, no. 4 (2018): 759.
2) Renal system
Harari, Florencia, Gerd Sallsten, Anders Christensson, Marinka Petkovic, Bo Hedblad, Niklas Forsgard, Olle Melander et al. "Blood Lead Levels and Decreased Kidney Function in a Population-Based Cohort." American Journal of Kidney Diseases (2018).
Lin, Ja-Liang, Dan-Tzu Lin-Tan, Kuang-Hung Hsu, and Chun-Chen Yu. "Environmental lead exposure and progression of chronic renal diseases in patients without diabetes." New England Journal of Medicine 348, no. 4 (2003): 277-286.
3) Hepatic system
Obeng-Gyasi, Emmanuel, Rodrigo X. Armijos, M. Margaret Weigel, Gabriel Filippelli, and M. Aaron Sayegh. "Hepatobiliary-Related Outcomes in US Adults Exposed to Lead." Environments 5, no. 4 (2018): 46.
Can, S., C. Bağci, M. Ozaslan, A. I. Bozkurt, B. Cengiz, E. A. Cakmak, R. Kocabaş, E. Karadağ, and M. Tarakçioğlu. "Occupational lead exposure effect on liver functions and biochemical parameters." Acta Physiologica Hungarica 95, no. 4 (2008): 395-403.
4) Reproductive system
Vigeh M., Smith D.R., Hsu P. How does lead induce male infertility? Iran. J. Reprod. Med. 2011;9:1–8.
Etc…. Also speak to its effects during the life course citing papers such as : Obeng-Gyasi, E., 2018. Lead Exposure and Oxidative Stress-A Life Course Approach in US Adults. Toxics, 6(3). Muller, C., Sampson, R.J. and Winter, A.S., 2018. Environmental Inequality: The Social Causes and Consequences of Lead Exposure. Annual Review of Sociology, (0). Reuben, A., Caspi, A., Belsky, D.W., Broadbent, J., Harrington, H., Sugden, K., Houts, R.M., Ramrakha, S., Poulton, R. and Moffitt, T.E., 2017. Association of childhood blood lead levels with cognitive function and socioeconomic status at age 38 years and with IQ change and socioeconomic mobility between childhood and adulthood. Jama, 317(12), pp.1244-1251. This will make the paper stronger and grab the reader’s attention. Methods: Did you calculate the required sample size prior to the study? What is the power of this study? Results and discussion: Please separate these sections. Conclusion: Please have a conclusion section:
Grashow, R., Sparrow, D., Hu, H. and Weisskopf, M.G., 2015. Cumulative lead exposure is associated with reduced olfactory recognition performance in elderly men: the Normative Aging Study. Neurotoxicology, 49, pp.158-164.
Author Response
Response to Reviewer 3 Comments
Point 1: Introduction: Talk about why we care about heavy metals bioaccumulation in plants as related to human health. Focus on lead since it is the most studied one. Specifically, speak to its effects on numerous organ systems such as the;
1) cardiovascular system
2) Renal system
3) Hepatic system
4) Reproductive system
Response 1: Thanks for your suggestion. This paper focused on the heavy metal accumulation in aquatic plants, which may indirectly do harm to human health. As a matter of fact, these aquatic plants are potentially useful to reduce the heavy metal in aquatic environment. Also this study will contribute to the further studies on the heavy metal accumulation capacity. We thought over about this carefully, and found it will be better to talk little about the physiological effect, because this may will mislead the readers. We thank you and hope your understanding. Thanks.
Point 2: Methods: Did you calculate the required sample size prior to the study? What is the power of this study?
Response 2: Following the common protocols of studies on heavy metal accumulation in aquatic plants, we chose adults plants as our typical plants. In the sampling season of October, these aquatic plants were in their thriving period, so the sample size are similar. In addition, the effects of sample size on aquatic plants is not the aim of this study. Thanks.
Point 3: Results and discussion: Please separate these sections.
Response 3: Thanks a lot for your suggestion. We though over about it and have tried to separate the results and discussion, but found it is better to keep it like this way if possible. Firstly, some results in this study, especially heavy metals in waters and sediments, have been exclusively reported in other study, integration will make the paper concise and tight. Secondly, we also learn from other papers published in International Journal of Environmental Research and Public Health, such as Niu et al., 2015, and feel it may be good to write in this way. If could, may we just insist to discuss in this manner. Of course, if you have any good opinions, please kindly let us know. Thanks again.
Point 4: Conclusion: Please have a conclusion section:
Response 4: Thanks, we have added the section of Conclusion.
Reviewer 4 Report
The present manuscript on accumulation of heavy metals in aquatic plants of the Taihu basin can be published. It does not have important methodological errors and although the information is local, it can also be considered general. However, the manuscript presents some issues that must be taken into consideration.
1) Objective iii does not seem applicable to this work. What are these factors and what research and results have been obtained? Clarify this aspect.
2) mL with a capital letter.
3) Lines 74-82. One aspect that in my opinion has not been taken into account is that the plants should have been initially washed with deionized water to remove metals on the surface or in the water that impregnates the surface. This is not indicated in the procedure.
4) Lines 93-98. How many replicates were used? It is necessary to indicate it, and it is an aspect that can be worrying in this type of work.
5) Some titles of the columns in the tables are cut, their appearance must be improved.
Author Response
Response to Reviewer 4 Comments
Point 1: Objective iii does not seem applicable to this work. What are these factors and what research and results have been obtained? Clarify this aspect.
Response 1: We mean the waters and sediments are the factors. Sorry for misunderstanding. We have corrected the Objective iii and also adjust the Objective I for better understanding. Thanks for advices.
Point 2: mL with a capital letter.
Response 2: Thanks, have been corrected.
Point 3: Lines 74-82. One aspect that in my opinion has not been taken into account is that the plants should have been initially washed with deionized water to remove metals on the surface or in the water that impregnates the surface. This is not indicated in the procedure.
Response 3: Thanks a lot for reminding. We stated in the part of sampling protocols and we have adjusted these sentences for better.
Point 4: Lines 93-98. How many replicates were used? It is necessary to indicate it, and it is an aspect that can be worrying in this type of work.
Response 4: We specified this in the paragraph of sampling protocols. Thanks.
Point 5: Some titles of the columns in the tables are cut, their appearance must be improved.
Response 5: Yes, they have been adjusted.
Round 2
Reviewer 1 Report
The paper has been improved following the suggestions. Now it can accepted for publication
Author Response
Response to Reviewer 1 Comments
Point 1: The paper has been improved following the suggestions. Now it can accepted for publication.
Response 1: Thanks very much for your valuable comments and supports on this manuscript.
Reviewer 2 Report
Dear authors
In view of the modificatons reported in your manuscript, the corrected version may be accepted for publication in the IJERPH journal.
Author Response
Response to Reviewer 2 Comments
Point 1: Dear authors, In view of the modificatons reported in your manuscript, the corrected version may be accepted for publication in the IJERPH journal.
Response 1: Thanks very much for your valuable comments and supports on this manuscript.
Reviewer 3 Report
Failed to address any of suggestions.
There is no evidence that the sample size is adequate to detect the effects the authors are claiming to have found. A “power analysis” is often used to determine sample size. Without demonstrating that the sample sizes used are adequate the entire paper is invalid.
The discussion and results section should be separated to make the article more readable. The results should be just that, the results, with the discussion comparing the what was found in the results with what is in literature and also discussing the limitations of the study. In its current form the paper is not as readable as it could be.
I chose reject largely due to the authors not clearly demonstrating that the sample size is adequate.
Author Response
Response to Reviewer 3 Comments
Failed to address any of suggestions.
There
is no evidence that the sample size is adequate to detect the effects
the authors are claiming to have found. A “power analysis” is often used
to determine sample size. Without demonstrating that the sample sizes
used are adequate the entire paper is invalid.
The discussion and
results section should be separated to make the article more readable.
The results should be just that, the results, with the discussion
comparing the what was found in the results with what is in literature
and also discussing the limitations of the study. In its current form
the paper is not as readable as it could be
I chose reject largely due to the authors not clearly demonstrating that the sample size is adequate.
Response : Dear Reviewer, we are so sorry that you may not be satisfied with our last responses. However, we still appreciate for your 4 suggestions for the MS, especially for reminding us to add a “Conclusion” which really help improve the manuscript a lot and we have added the “Conclusion” section in last revision following your suggestions. We are not sure it’s because you haven’t paid attention to that or you have other suggestions on conclusion.
For the point 1 of your suggestion, we really look forward to your understanding on our explanation that talking about the physiological effects of heavy metal especially only talking about Pb in this manuscript will make readers confused. If possible, could we humbly just keep the Introduction as it is.
For Point 2, we really confused about your suggestions on sample size, because the aquatic plants were all in their thriving period, it seems very hard to distinguish them by sample size.
We appreciate your kindly helps and comments on the manuscript, and hope to get your understandings.
Reviewer 4 Report
The manuscript has been revised correctly and the answers to the questions raised have been convincing, therefore the manuscript can be published.
Author Response
Response to Reviewer 4 Comments
Point 1: The manuscript has been revised correctly and the answers to the questions raised have been convincing, therefore the manuscript can be published.
Response 1: Thanks very much for your valuable comments and supports on this manuscript.